# Identification of MicroRNAs Associated with Histological Grade in Early-Stage Invasive Breast Cancer

**DOI:** 10.3390/ijms25010035

**Published:** 2023-12-19

**Authors:** Sasagu Kurozumi, Naohiko Seki, Eriko Narusawa, Chikako Honda, Shoko Tokuda, Yuko Nakazawa, Takehiko Yokobori, Ayaka Katayama, Nigel P. Mongan, Emad A. Rakha, Tetsunari Oyama, Takaaki Fujii, Ken Shirabe, Jun Horiguchi

**Affiliations:** 1Department of Breast Surgery, International University of Health and Welfare, Chiba 286-8520, Japan; 2Department of General Surgical Science, Gunma University Graduate School of Medicine, Gunma 371-8511, Japanftakaaki@gunma-u.ac.jp (T.F.);; 3Department of Functional Genomics, Chiba University Graduate School of Medicine, Chiba 260-8670, Japan; naoseki@faculty.chiba-u.jp; 4Initiative for Advanced Research, Gunma University, Gunma 371-8511, Japan; 5Department of Diagnostic Pathology, Gunma University Graduate School of Medicine, Gunma 371-8511, Japanoyama@gunma-u.ac.jp (T.O.); 6Biodiscovery Institute, Faculty of Medicine and Health Sciences, University of Nottingham, Nottingham NG7 2RD, UK; 7Academic Unit for Translational Medical Sciences, School of Medicine, University of Nottingham, Nottingham NG7 2RD, UK; 8Pathology Department, Hamad Medical Corporation, Doha P.O. Box 3050, Qatar

**Keywords:** invasive breast cancer, histological grade, microRNA

## Abstract

This study aimed to identify microRNAs associated with histological grade using comprehensive microRNA analysis data obtained by next-generation sequencing from early-stage invasive breast cancer. RNA-seq data from normal breast and breast cancer samples were compared to identify candidate microRNAs with differential expression using bioinformatics. A total of 108 microRNAs were significantly differentially expressed in normal breast and breast cancer tissues. Using clinicopathological information and microRNA sequencing data of 430 patients with breast cancer from The Cancer Genome Atlas (TCGA), the differences in candidate microRNAs between low- and high-grade tumors were identified. Comparing the expression of the 108 microRNAs between low- and high-grade cases, 25 and 18 microRNAs were significantly upregulated and downregulated, respectively, in high-grade cases. Clustering analysis of the TCGA cohort using these 43 microRNAs identified two groups strongly predictive of histological grade. miR-3677 is a microRNA upregulated in high-grade breast cancer. The outcome analysis revealed that patients with high miR-3677 expression had significantly worse prognosis than those with low miR-3677 expression. This study shows that microRNAs are associated with histological grade in early-stage invasive breast cancer. These findings contribute to the elucidation of a new mechanism of breast cancer growth regulated by specific microRNAs.

## 1. Introduction

With the recent development of diagnostic and therapeutic techniques, the prognosis of patients with early-stage breast cancer has greatly improved. However, approximately 20% of patients with early-stage disease still suffer from recurrences [1]. Therefore, it is necessary to refine the prognostic and predictive classification of breast cancer.

Histological grade, which is one of the main prognostic factors in breast cancer [2,3], is generally defined on a scale comprising three grades based on three morphological features: nuclear atypia/pleomorphism (morphological characteristics of individual cancer cells), tubule formation (morphological characteristics that reflect the ability of cancer cells to form tubules similar to normal tissue), and mitotic activity (proliferative potential of cancer cells). The Nottingham Prognostic Index, which is a widely used prognostic tool for patients with breast cancer, is calculated using equal prognostic weights of histological grade and lymph node stage. Histological grade was included in the recent AJCC prognostic staging system for breast cancer. Several previous studies also demonstrated that this histological grade classification is effective in predicting chemotherapy response and prognosis in breast cancer patients [4,5]. However, the molecular biological mechanism defining the histological grade remains unclear. Therefore, translational research is needed to elucidate the molecular mechanisms underlying such morphological features and identify genes predictive of histological grade.

MicroRNAs (miRNAs) are single-stranded non-protein-coding RNAs that function as critical post-transcriptional gene regulators. miRNAs negatively regulate gene expression by binding to their selective messenger RNAs, leading to translational repression [6]. Aberrant expression of miRNAs has been associated with the development of various human diseases, including breast cancer. Although some individual miRNAs appeared overexpressed in high-grade breast cancer, miRNA expression profiling in relation to histological grading remains unclear. Discovering miRNAs related to the histological grade of breast cancer that defines the proliferative and metastatic potentials can help not only discover prognostic factors but also develop new early diagnostic tools for patients with early-stage breast cancer and new therapeutic targets.

Thus, this study aimed to identify miRNAs differentially expressed in breast tumors depending on their histological grade using comprehensive miRNA analysis data obtained by next-generation sequencing from early-stage breast cancer.

## 2. Results

### 2.1. Identification of miRNAs Associated with Breast Cancer

Several miRNAs are involved in the transformation of normal cells into cancer cells. Differences in miRNAs were determined between normal breast tissues and cancer cells to identify miRNA that are likely to be involved in the carcinogenetic process. The comparison of the RNA-seq data between normal breast and breast cancer tissues using the edgeR method showed that 108 miRNAs were significantly differentially expressed (false discovery rate [FDR] < 0.05). Appendix A shows the list of these 108 miRNAs.

### 2.2. Clinicopathological and Prognostic Significance of Histological Grade in the Cancer Genome Atlas (TCGA) Cohort

Table 1 shows the clinicopathological characteristics of the TCGA cohort examined in this study, and Figure 1 shows the distribution of the histological grade. No patient received preoperative chemotherapy. Of the 430 breast cancer cases in the TCGA cohort, 256 (59.5%) were of histological grades 1 and 2, and 174 (40.5%) were of histological grade 3. In this cohort, grade 3 was significantly associated with lymph node metastasis (*p* = 0.011) and estrogen receptor (ER) negativity (*p* < 0.0001; Appendix A). Patients with grade 3 breast cancer tended to have worse survival rates than those with grade 1 and 2 breast cancer. However, the difference was not statistically significant (hazard ratio [HR] 1.58, 95% confidence interval [CI] 0.96–2.59, *p* = 0.072). 

### 2.3. miRNAs Associated with Histological Grade

The expression of the 108 miRNAs that were differentially expressed between normal and cancerous tissues was compared between low-grade and high-grade breast cancer cases. In high-grade cases, 25 and 18 miRNAs were significantly upregulated and downregulated, respectively (FDR < 0.05; Table 2).

Hierarchical clustering analysis of the 430 TCGA cases using the 43 miRNAs classified the cohort into the following two subgroups: subgroup 1 (*n* = 201 cases, 46.7%) and subgroup 2 (*n* = 229 cases, 53.3%). Figure 2 shows a dendrogram classifying the 430 cases. Histological grade 3 was significantly more prevalent in subgroup 2 than in subgroup 1 (*p* < 0.0001). Additionally, subgroup 2 had a higher frequency of ER-negative cases (*p* < 0.0001; Appendix A). These 43 miRNAs were significantly associated with the “protein processing in endoplasmic reticulum (hsa04141)” KEGG pathway (*p* < 0.0001). Key miRNAs involved in this pathway included the highly novel miR3677. Furthermore, miR3677 was found to be involved in this pathway via a single protein, ribophorin II (RPN2).

### 2.4. Clinicopathological and Prognostic Significance of miR-3677

miR-3677 is one of the top miRNAs involved in defining the histological grade and has been reported to play a key role in other cancers. Thus, the relationship between miR-3677 and clinicopathological features and histological grade was further investigated in the TCGA cohort. miR-3677 expression was significantly positively associated with histological grade (Figure 3a). High miR-3677 expression was significantly associated with other features characteristic of aggressive behavior, including ER negativity (*p* = 0.00011) and HER2 positivity (*p* = 0.00012; Appendix A).

The outcome analysis demonstrated that patients with high miR-3677 expression had a significantly worse prognosis than those with low miR-3677 expression (HR 2.20, 95% CI 1.28–3.76, *p* = 0.0042; Figure 3b). Furthermore, multivariate analysis revealed that high miR-3677 expression was an independent predictor of poor prognosis (HR 2.45, 95% CI 1.28–4.69, *p* = 0.0068; Table 3).

## 3. Discussion

In this study, the proprietary miRNA dataset from our facility and the TCGA miRNA database were used to identify 43 miRNAs that were significantly correlated with breast cancer histological grade.

Histological grade determination is an efficient and cost-effective method for predicting the proliferation and metastatic potential of early-stage breast cancer. In a large-scale study conducted by Henson et al. [7] that evaluated the survival rate of 22,616 breast cancer cases, the survival rate of histological grade 1 stage II patients was the same as that of grade 3 stage I patients. Furthermore, despite having LN positivity, grade 1 patients with a tumor size of <2 cm had a 5-year survival rate of 99% and good prognoses. These results are consistent with those of a previous study conducted by Rakha et al. [8], utilizing data from 2219 operable breast cancer patients with long-term follow-up and demonstrating that histological grade is a key determinant of breast cancer outcome. These results suggest that histological grade 1 and 3 breast cancers have different molecular profiles and highlight the importance of identifying the set of genes predictive of the histological grade. Sotiriou et al. [9] developed a panel of 97 genes to differentiate between grade 1 and 3 breast cancer cases using the Nottingham grading system. Comparing the prognosis and predictive accuracy of gene signatures based on this gene panel with those of Oncotype DX^®^, it was found that both methods demonstrated similar differences in the distant metastasis-free survival rate between the low- and the high-risk groups [10,11]. These studies shed some light on the differences between the molecular profiles of tumors with different histological grades at the genomic, transcriptomic, and proteomic levels.

Recently, the clinical utility of the histological grade in the selection of drug therapy for early-stage breast cancer has increased. Chemotherapy targets molecular pathways related to cell proliferation and affects biomarkers associated with cell proliferation, including histological grade. Although invasive breast cancer of histological grade 3 has a poor prognosis, chemotherapy is highly efficient for treating this type of cancer. Histological grade before neoadjuvant chemotherapy has been reported to be associated with a pathological complete response rate after neoadjuvant treatment. Histological grade is a predictor of response to preoperative trastuzumab combination chemotherapy in patients with HER2-positive breast cancer [12,13]. According to the St. Gallen guidelines [14], individual treatment decision is recommended based on clinicopathological factors, such as tumor size, lymph node metastasis status, histological grade, Ki67 labeling index, quantitative expression of hormone receptors, and genomic signature, for stage 2 breast cancer patients when selecting chemotherapy for ER-positive–HER2-negative early-invasive breast cancer. Cyclin-dependent kinase inhibitors, including palbociclib, ribociclib, and abemaciclib, target factors related to the cell cycle, and their utility has already been confirmed in clinical trials [15,16,17]. The monarchE trial demonstrated that abemaciclib improved the prognosis in patients with high-risk ER-positive and HER2-negative early-stage breast cancer. In this trial, “high risk” was defined as “patients with four or more positive pathologic axillary lymph nodes or 1–3 positive axillary lymph nodes and at least one of the following: tumor size > 5 cm, histological grade 3, or centrally assessed Ki-67 expression > 20%” [18].

miRNAs comprise small noncoding RNA molecules that regulate gene expression post-transcriptionally and are involved in the onset, progression, and metastasis of breast cancer [19,20]. Recent studies reported that a diagnostic miRNA signature can be employed to assess tumor aggressiveness, drug response, and outcomes in patients with breast cancer [19,21,22]. Furthermore, miRNA expression profiles have been reported to be different for each molecular subtype of breast cancer [6]. Liquid biopsy using a blood sample for collecting data is equivalent to the method that uses a solid tissue sample [23]. Substances and materials produced by tumors, such as carcinoembryonic antigen, carbohydrate antigen 19-9, circulating tumor cells, cell-free DNA/circulating tumor DNA, and miRNA, can be measured by liquid biopsy [24]. Liquid biopsy enables the monitoring of circulating tumor cells throughout the body via the blood and their biological features, such as genes originating from such tumor cells [25]. Since the molecular features of biological substances can change in a short period of time, this method can provide the real-time identification of factors directly contributing to tumor proliferation and metastasis. However, the scientific stage where all the data can be obtained by liquid biopsy alone has not been reached, and histopathological evaluations by pathologists are still crucial in clinical practice. Few studies have confirmed the clinical utility of liquid biopsies for monitoring serum miRNA fluctuations. Newly discovered miRNAs can provide a strategically novel option for cancer diagnosis and treatment by determining the histological grade of breast cancer.

This study focused on one of the miRNAs, miR-3677, because of its novelty and significance. This miRNA has been reported to be a significant prognostic predictor for hepatocellular carcinoma [26]. Peng et al. [27] compared the expression levels of miR-3677 in noncancerous and breast cancer cells and showed significantly higher miR-3677 levels in breast cancer cells. Furthermore, they reported that the overexpression of miR-3677 promoted the proliferation and colony formation of breast cancer cells. Additionally, they performed a bioinformatic analysis using data obtained from the TCGA database and identified transducin-like enhancer protein 3 (TLE3) as a potential target of miR-3677 [27]. TLE3 is a full-length human TLE family member that plays an important role in breast cancer cell differentiation and proliferation [28,29]. Moreover, patients with breast and ovarian cancers with high TLE3 expression were reported to respond well to a chemotherapeutic regimen that included taxane [30,31]. The expression levels of miR-3677 were reported to be negatively correlated with TLE3 expression in breast cancer tissue, suggesting that miR-3677 is involved in breast cancer cell proliferation and metastasis by suppressing TLE3 expression. This study demonstrated that miR-3677 expression was significantly correlated with histological grade. Therefore, it may be a strong prognostic predictor. The pathway analysis revealed that miR-3677 was associated with the “protein processing in the endoplasmic reticulum (hsa04141)” KEGG pathway via a single protein, RPN2, which is a membrane protein that localizes to the rough endoplasmic reticulum. Based on gene expression profiles of breast cancer, Ochiya et al. [32] revealed that RPN2 was involved in the treatment resistance of breast cancer and found that RPN2 was specifically expressed in ESA+/CD44+/CD24− breast cancer cells, which are considered “breast cancer stem cells”. Further studies are needed to analyze the molecular mechanisms involved in regulating miR-3677 and RPN2 expression using in vitro and in vivo assays.

## 4. Materials and Methods

### 4.1. Discovery Cohort

A total of 20 surgically excised tissue samples were collected at Gunma University Hospital to construct miRNA expression signatures of human breast cancer, including 5 normal breast, 5 ER-positive/HER2-negative breast cancer, 5 HER2-positive breast cancer, and 5 triple-negative breast cancer tissue samples. The clinicopathological characteristics of the 15 breast cancer cases are shown in Appendix A. Total RNA was isolated from these frozen samples using the TRIzol reagent (Invitrogen, Carlsbad, CA, USA) according to the manufacturer’s instructions. The RNA extraction method using this reagent is described on the company’s website (https://www.thermofisher.com/, accessed on 20 November 2023). The miRNA expression signatures of the 20 samples were generated by small RNA sequencing using HiSeq 2500 (Illumina, San Diego, CA, USA). Quality control and normalization were performed according to previous analyses [33,34]. Clean reads were obtained after removing adaptor sequences, reads having greater than 10% of unknown bases, and reads with a proportion of low-quality bases (base with quality value ≤ 5) greater than 10%. These miRNA data were registered in Gene Expression Omnibus (GSE248460). A comparison between the 5 noncancerous and the 15 cancerous samples was performed, and miRNAs with different expression were identified using the edgeR method (Figure 4).

### 4.2. TCGA Validation Cohort

A total of 430 patients with breast cancer were enrolled from the TCGA, with clinicopathological information and miRNA data measured by RNA-seq downloaded from the Genomic Data Commons Data Portal and cBioPortal [35,36]. Differences in miRNA expression between low-grade and high-grade breast cancers were determined using the edgeR method [37], and FDR < 0.05 was considered significant (Figure 1). The MA plot of the miRNAs is shown in Appendix A. The TCGA data were log2-transformed and normalized to the provided miRNA expression data before cluster analysis. Cluster analysis and heatmap generation were performed using Cluster 3.0 and Java Treeview [38]. Pathway analysis was performed using mirPath v.3 (https://dianalab.e-ce.uth.gr/html/mirpathv3/index.php?r=mirpath, accessed on 12 March 2023) to calculate the pathways that were significantly associated with the candidate miRNAs in the KEGG analysis [39]. 

### 4.3. Statistical Analysis

The association between histological grade and candidate miRNAs was assessed using the Mann–Whitney test, whereas the association between candidate miRNAs and other clinicopathological factors was assessed using the chi-square test. Survival analysis was performed using the Kaplan–Meier method to illustrate the survival curve, and HR and 95% CI were calculated using the Cox proportional hazards regression model. A *p*-value < 0.05 was considered significant. Statistical analyses were performed using SPSS Statistics version 24.0 (IBM Corp, Armonk, NY, USA) and GraphPad Prism 7.03 (GraphPad Software Inc., La Jolla, CA, USA).

## 5. Conclusions

In this study, a miRNA profile associated with histological grade was found. The study results may lead to the discovery of new miRNAs that may become candidates for new RNA-targeted therapies. However, research clarifying the mechanisms by which miRNA molecules influence the histological grade using comprehensive expression data obtained by next-generation sequencing for breast cancer is insufficient. Thus, further validation research and functional analysis including real-time polymerase chain reaction assays of the miRNAs related to histological grade are necessary to use them in breast cancer diagnosis and treatment.

## Figures and Tables

**Figure 1 ijms-25-00035-f001:**
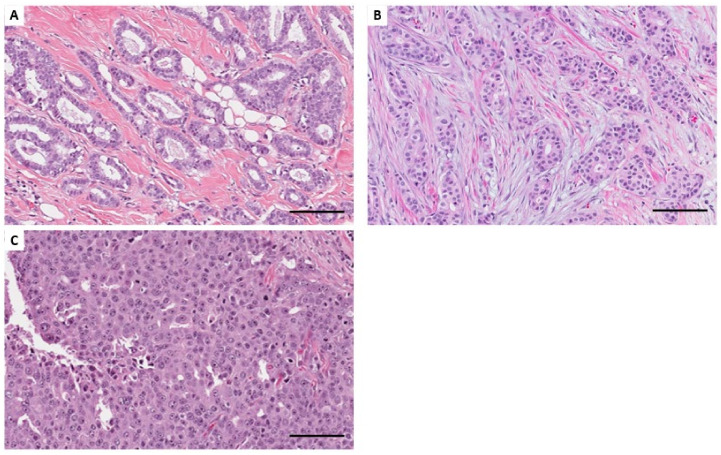
Histological appearance defining the histological grade. (**A**) Grade 1: well-differentiated tumor with tubule formation (>75%), a mild degree of nuclear pleomorphism, and a low mitotic count. (**B**) Grade 2: moderately differentiated tumor. (**C**) Grade 3: poorly differentiated tumor with a marked degree of cellular pleomorphism, frequent mitoses, and almost no tubule formation (<10%) (HE, scale bar = 100 μm).

**Figure 2 ijms-25-00035-f002:**
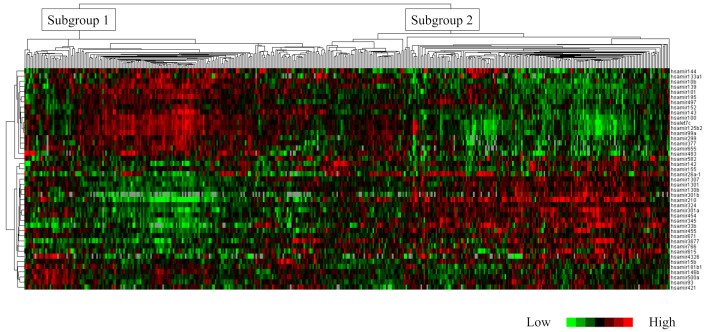
Clustering analysis of 43 microRNAs (miRNAs) involved in histological grading. With grade changes, the expression pattern of candidate miRNAs also clearly changes.

**Figure 3 ijms-25-00035-f003:**
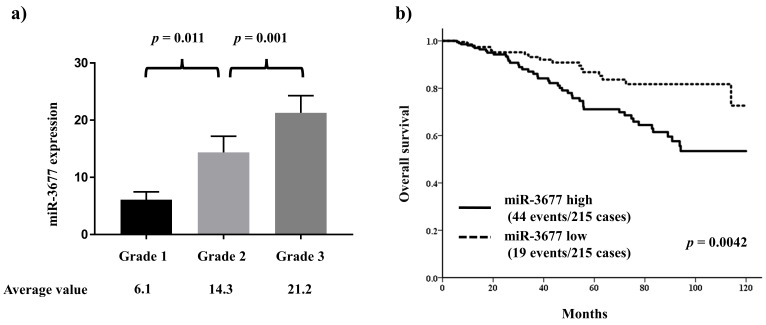
Significance of miR-3677 in breast cancer. (**a**) In 430 patients with breast cancer, a significant correlation was observed between miR-3677 and histological grade. (**b**) miR-3677 is a strong prognostic factor.

**Figure 4 ijms-25-00035-f004:**
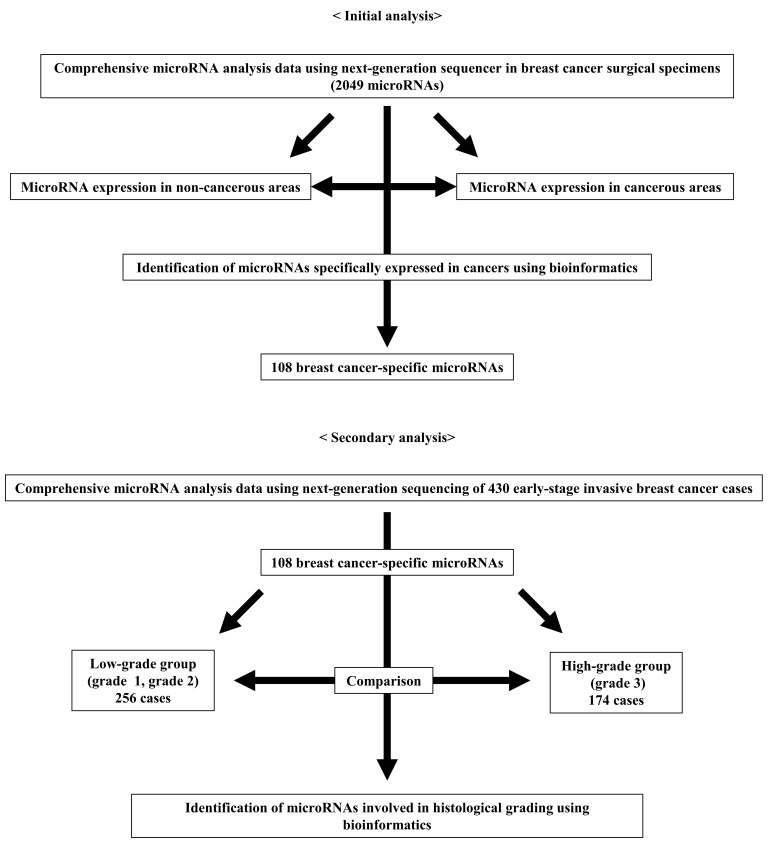
Methods for identifying miRNAs involved in defining breast cancer histological grade using bioinformatics. Initially, 108 miRNAs with significantly altered expression in tumor areas compared with non-tumor areas were extracted from our hospital’s exhaustive miRNA data (2049 miRNAs). Next, the expression levels of these 108 miRNAs from TCGA were compared between the low and high histological grade groups.

**Table 1 ijms-25-00035-t001:** Clinicopathological characteristics of the TCGA cohort in this study.

Characteristics	*n*
Age range, years	
24–40	31
41–59	212
60 and over	187
Tumor size	
<2.0 cm	115
≥2.0 cm	315
Nodal status	
Negative	200
Positive	226
Unknown	4
Histological grade	
Grade 1	58
Grade 2	198
Grade 3	174
Estrogen receptor status	
Positive	326
Negative	96
Unknown	8
HER2 status	
Positive	69
Negative	298
Unknown	63

**Table 2 ijms-25-00035-t002:** List of 43 microRNAs (miRNAs) significantly associated with breast cancer of histological grade 3.

Upregulated miRNAs	logFC	FDR	Downregulated miRNAs	logFC	FDR
hsamir210	1.09	1.74 × 10^−29^	hsamir483	−1.35	7.91 × 10^−22^
hsamir301b	1.56	1.25 × 10^−22^	hsamir195	−0.81	1.60 × 10^−11^
hsamir455	0.98	2.80 × 10^−19^	hsamir101	−0.74	1.21 × 10^−10^
hsamir301a	0.88	2.94 × 10^−19^	hsamir10b	−0.91	2.15 × 10^−10^
hsamir130b	0.85	6.90 × 10^−13^	hsamir139	−0.69	5.26 × 10^−9^
hsamir142	0.67	8.42 × 10^−9^	hsamir99a	−0.61	7.09 × 10^−9^
hsamir1307	0.59	2.75 × 10^−8^	hsalet7c	−0.51	1.37 × 10^−7^
hsamir345	0.66	8.95 × 10^−8^	hsamir100	−0.57	4.59 × 10^−6^
hsamir454	0.55	8.95 × 10^−8^	hsamir143	−0.50	1.23 × 10^−5^
hsamir500a	0.61	1.30 × 10^−7^	hsamir377	−0.54	0.00027
hsamir93	0.57	6.01 × 10^−7^	hsamir125b2	−0.26	0.00069
hsamir155	0.55	2.59 × 10^−6^	hsamir152	−0.42	0.0010
hsamir4326	0.53	2.92 × 10^−6^	hsamir655	−0.46	0.0081
hsamir324	0.54	4.89 × 10^−6^	hsamir133a1	−0.47	0.0094
hsamir1301	0.49	1.32 × 10^−5^	hsamir497	−0.34	0.0098
hsamir766	0.53	6.16 × 10^−5^	hsamir299	−0.35	0.019
hsamir421	0.62	0.00013	hsamir144	−0.16	0.020
hsamir181b1	0.36	0.0016	hsamir26a-1	−0.37	0.047
hsamir3677	0.34	0.0050			
hsamir671	0.34	0.0073			
hsamir33b	0.24	0.0084			
hsamir15b	0.30	0.0085			
hsamir146b	0.28	0.012			
hsamir582	0.25	0.014			
hsamir615	0.15	0.035			

**Table 3 ijms-25-00035-t003:** Survival analysis of clinicopathological factors, including miR-3677.

Factors		Univariate Analysis			Multivariate Analysis		
		HR	95% CI	*p*-Value	HR	95% CI	*p*-Value
miR-3677 expression	Low	Reference			Reference		
	High	2.20	1.28–3.76	0.0042	2.45	1.28–4.69	0.0068
Tumor size	pT1	Reference			Reference		
	pT2–pT4	1.49	0.85–2.64	0.17	0.92	0.49–1.72	0.80
Nodal status	Negative	Reference			Reference		
	Positive	2.33	1.34–4.04	0.0026	2.42	1.28–4.59	0.0068
ER	Negative	Reference			Reference		
	Positive	0.47	0.28–0.81	0.0060	0.55	0.30–1.01	0.055
HER2	Negative	Reference			Reference		
	Positive	2.29	1.27–4.13	0.0062	1.30	0.67–2.53	0.43

ER: estrogen receptor; HER2: human epidermal growth factor 2; HR: hazard ratio; CI: confidence interval.

## Data Availability

The datasets generated and/or analyzed during the current study are not publicly available due to the regulation of the Institutional Review Board of Gunma University but are available from the corresponding author upon reasonable request.

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
