# Peer review of "Identification of MicroRNAs Associated with Histological Grade in Early-Stage Invasive Breast Cancer"

_ijms, 2023, doi:10.3390/ijms25010035_

Round 1

Reviewer 1 Report (New Reviewer)

Comments and Suggestions for Authors

The manuscript by Kurozumi S. et al.., entitled Identification of MicroRNAs Associated with Histological Grade in Early-Stage Invasive Breast Cancer, represents an interesting and useful study regarding new mechanism of breast cancer growth by the specific microRNAs regulation. The authors demonstrated a relationship between microRNAs and histological grade in invasive early-stage breast cancer. The study proved that patients with high miR-3677 expression had a considerably worse prognosis than those with low miR-3677 expression.

The manuscript is valuable and well-structured, and the results and discussion sections are very well developed.

In the materials and methods section, the authors didn’t present the complete methodology in some places. For example, according to the manufacturer’s instructions or according to previous analyses. Would be interesting to briefly underline the most important steps.

Author Response

Thank you for your valuable comment. TRIzol reagents are common reagents used for RNA extraction, and are used by all researchers studying molecular biology. A website with the details of experimental methods for this reagent is listed. And, we have registered our RNAseq data with GEO. The GSE number was added in the Methods as follows;

“Total RNA was isolated from these frozen samples using TRIzol reagent (Invitrogen, Carlsbad, CA, USA) according to the manufacturer’s instructions. The RNA extraction method using this reagent is described on the company's website (https://www.thermofisher.com/). The miRNA expression signatures of the 20 samples were generated by small RNA sequencing using HiSeq 2500 (Illumina, San Diego, CA, USA). Quality control and normalization were performed according to previous analyses [33,34]. Clean reads were obtained after removing adaptor sequences, reads having greater than 10% of unknown bases, and reads with low-quality bases (base with quality value ≤ 5) greater than 10% in a read. This miRNA data was registered in Gene Expression Omnibus (GSE248460).”

Reviewer 2 Report (New Reviewer)

Comments and Suggestions for Authors

In this study, the authors performed a bioinformatic analysis using miRNA dataset obtained from their facility and TCGA database to identify microRNAs associated with histological grade in breast cancer. The authors found 43 miRNAs that were significantly correlated with histological grade. Among them, the expression of miR-3677 in tumor tissue is negatively correlated with histological grade and patient survival time, and can be used as a prognostic marker for breast cancer.

In general, the interpretation of the data is reasonable, and the illustration is presented in a good quality. However, the oncogenic role of miR-3677 in breast cancer has been confirmed in previous studies. Although this may echo the findings of this study, it also reduces its innovativeness.

Moreover, although the results of this study are helpful for the prognosis and treatment of breast cancer, however, it is a pity that this study only presents the results of biological information analysis, and most of the results come from the TCGA public database. The authors should verify the correlation between miR-3677 expression and histological grade in specimens of at least 50 breast cancer patients to support their analysis results. In addition, the expression of miR-3677 in serum can also be detected, which can increase the innovation and clinical application value of this study.

Comments on the Quality of English Language

The manuscript contains many grammatical and syntax errors and must be carefully edited.

Author Response

Thank you for your constructive comment. As you mentioned, it is important to verify the present results using our own cohort. But, unfortionately, we have no cohort to verify them and evaluate serum miR-3677. We have added the description in term of the importance of validation analysis for miR-3677 in the Conclusion part as follows;

“In this study, a miRNA profile associated with histological grade was found. The study results may lead to the discovery of new miRNAs that may become candidates for new RNA-targeted therapies. However, research clarifying the mechanisms of miRNA molecules on histological grade using comprehensive expression data obtained by next-generation sequencing for breast cancer is insufficient. Thus, further validation research and functional analysis including real-time polymerase chain reaction assay of the miRNAs related to histological grade is necessary to use them in breast cancer diagnosis and treatment.”

And, we have consulted a co-author who is a native English speaker to read though our manuscript again and check the grammatical error carefuly. The revised manuscript is re-submitted.

Round 2

Reviewer 2 Report (New Reviewer)

Comments and Suggestions for Authors

Although the authors were unable to verify their findings in clinical specimens, they have addressed their shortcomings in the Discussion section. Therefore, I consider the revised manuscript acceptable.

This manuscript is a resubmission of an earlier submission. The following is a list of the peer review reports and author responses from that submission.

Round 1

Reviewer 1 Report

Comments and Suggestions for Authors

In this manuscript, Kurozumi and collaborators focus on the association of microRNAs and histological grade in breast cancer patients. They performed RNAseq analysis in 5 normal breasts and 15 breast cancer tissues (5 ER/PR positive, 5 HER2 positive, and 5 triple negative) and found 108 significantly differentially expressed between normal breast and breast cancer tissue. They then accessed the expression of these miRNAs in the TCGA database and described in more detail the association of miR-3677 expression with histologic grade and overall survival.

The study of early-stage breast cancer is important to shed light on mechanisms that can be used to improve patient outcomes. However, the experimental design of the study is critical to address this subject and the presentation of the data is key to support the results and there are major concerns in this manuscript. Regarding the experimental design in this study, it is not clear why the selection of the 108 miRNAs was based on a comparison between normal breast and breast cancer tissues. Why not investigate the differentially expressed miRNAs between low histological grade and high histological grade? Another aspect that was lacking is the description of how the samples used for the RNAseq analysis were selected and classified. There are no data showing the staging of these samples. This study is also heavily based on the analysis of TCGA data and does not explore the RNAseq data generated by the authors. There is no evidence of the differentially expressed profile of these 108 microRNAs, only a table with their names. It is critical that these data are shown through graphs, such as a heatmap, and a supplementary table showing the raw data. In this regard, the authors also have not deposited the RNAseq data as instructed by the journal guidelines “New high throughput sequencing (HTS) datasets (RNA-seq, ChIP-Seq, degradome analysis, …) must be deposited either in the GEO database or in the NCBI’s Sequence Read Archive (SRA).”.

Another major issue is that there are several overstatements that are present in the text which are not substantiated by the data provided. One example is that although the authors state in the title that they are investigating early-stage invasive breast cancer, they do not present any data supporting this statement. Staging has not been described by any of the cohorts studied. Another example is the discussion on liquid biopsy, in the last paragraph of the discussion. The authors did not show any data with blood analysis and this statement is off-topic. Lastly, the conclusion has several affirmations that go beyond the scope of this study. There are no supporting data in this manuscript showing the relationship among microRNAs, histopathological grade, drug treatment, and drug resistance. It also does not show the discovery of new miRNAs. The discussion cited at least 2 studies (ref#23 and ref#24) that described the role of miR-3677 in cancer. Based on all these issues, I do not believe this paper is suitable for publication.

Comments on the Quality of English Language

The English was appropriated and just requires minor editing. 

Reviewer 2 Report

Comments and Suggestions for Authors

Kurozumi et al manuscript utilized the comparison between histological grade and miRNAs expression to identify possible candidates for prognosis in early stage breast cancer. They identified miRNA 3677 as a promising candidate as its expression was correlated with histological grade as well as survival. The manuscript has interesting and relevant findings. Importantly, the role of miRNA 3677 in breast cancer is poorly explored. Therefore, the finding that it was correlated with survival is novel and very important for the oncology field. Nevertheless, few points were raised:

1) Table 2 could show not only which miRNAs are up and down regulated but also the fold change of expression. It would contribute to clarify which miRNA expression are the most altered, including miRNA 3677.

2) The data with miRNA 3677 is poorly explored. Despite the interesting findings correlated with histological grade and survival, I wonder if it is possible to correlate this miRNA expression with other clinical data, such as tumor size, lymph node spread, metastasis, etc.

3) Please mention the findings with miRNA 3677 in the abstract as well as in the title.

4) line 112-113: please correct the figure legend

Reviewer 3 Report

Comments and Suggestions for Authors

Regarding the manuscript entitled “Identification of microRNAs Associated with Histological Grade in Early-stage Invasive Breast Cancer”, the authors aimed to identify microRNAs involved in histological grade using comprehensive microRNA analysis data utilizing next-generation sequencing in early-stage invasive breast cancer. They compared the RNA-seq data from normal breast and breast cancer samples to identify candidate microRNAs with different expressions using bioinformatics.

The study is interesting. However, major revision is needed before its acceptance.

Please use “histopathological” in replacing “histological” because you deal with breast cancer which is a pathological case.

Please add a figure showing the histopathology evaluation of normal and the three grades of breast cancer.

Please avoid using “we” in the whole manuscript.

Please update your citation and references list to 2021-2023.

Reviewer 4 Report

Comments and Suggestions for Authors

In the manuscript " Identification of microRNAs Associated with Histological Grade in Early-stage Invasive Breast Cancer", submitted by Kurozumi et al., the authors took an omics approach and identified that microRNAs are associated with histological grade in early-stage invasive breast cancer.

As a researcher working with miRNAs, I found substantial innovation in the approach to developing the data and its usefulness for a wide range of readers. The manuscript may be recommended for publication in its present form. Nevertheless, the authors may be asked to look into the following minor editorial corrections:

·      The author must be careful of the manuscript's spelling and other grammatical errors.

·      The volcano plot should be reported for the DE miRs.

·      Table 2 should be improved.

·      Figure quality should be improved.

·      The discussion part should be stronger.

Comments on the Quality of English Language

The author must be careful of the manuscript's spelling and other grammatical errors.